# Efficacy, tolerability, and safety of an innovative medical device for improving oral accessibility during oral examination in special-needs patients: A multicentric clinical trial

Mathieu Mogenot[1], Laurence Hein-Halbgewachs[1], Christophe Goetz[2], Nadia Ouamara[2], Dominique Droz-Desprez[3], Catherine Strazielle[3], Sylvie Albecker[4], Brigitte Mengus[4], Marion Strub[5], Marie-Cécile Manière[5], Pascal Richardin[6], Stéphane Wang[6], Giuseppa Piga[2], Amélie Dalstein[7], Daniel Anastasio[1]*

1 Department of Odontology, Metz-Thionville Regional Hospital, Bel-Air Hospital, Thionville, France,
2 Clinical Research Support Unit, Metz-Thionville Regional Hospital, Mercy Hospital, Metz, France,
3 Department of Odontology, Nancy Regional University Hospital, Nancy, France, 4 Handident Alsace Network, St François Clinic, Haguenau, France, 5 Department of Odontology, Strasbourg Regional University Hospital, Strasbourg, France, 6 Department of Odontology, Metz-Thionville Regional Hospital, Mercy Hospital, Metz, France, 7 Department of Odontology, Emile-Durkenheim Hospital, Epinal, France

* projet-recherche@chr-metz-thionville.fr

**Data Availability Statement:** Data cannot be shared publicly because of confidentiality

## Abstract

### Background

People with special needs have high unmet oral healthcare needs, partly because dentists find it difficult to access their oral cavity. The Oral Accessibility Spatula aims to improve oral accessibility. This prospective multicenter interventional open-label non-randomized patient-self-controlled trial assessed the ability of the spatula to improve the oral accessibility of special-needs patients during dental examinations.

### Methods

The cohort was a convenience sample of minor and adult patients with special needs due to physical, intellectual, and/or behavioral disorders who underwent dental check-up/treatment in five French tertiary hospitals/private clinics in 2016–2018 and evinced some (Venham-Score = 2–4) but not complete (Venham-Score = 5) resistance to oral examination. After inclusion, patients underwent oral examination without the spatula and then immediately thereafter oral examination with the spatula. Primary outcome was Oral Accessibility Score (0–12 points; higher scores indicate visualization and probing of the tooth sectors). Secondary outcomes were patient toleration (change in Venham-Score relative to first examination), safety, and Examiner Satisfaction Score (0–10; low scores indicate unsatisfactory examination).

concerns. Data are available from the Clinical
Research Support Unit from the Metz-Thionville
Regional Hospital (Plateforme d'Appui à la
Recherche Clinique du CHR de Metz-Thionville) for
researchers who meet the criteria for access to
confidential data. Data requests may be sent to the
corresponding author or to the data protection
offficer of our hospital: dpo@chr-metz-thionville.fr.

**Funding:** The study was funded by the DGOS
(Direction Générale de l'Offre de Soin = Directorate
of Health Care Supply), a French government
agency. The funders did not play any role in study
design, data collection and analysis, decision to
publish or preparation of the manuscript.

**Competing interests:** The authors have declared
that no competing interests exist.

## Results

The 201 patients were mostly non-elderly adults (18–64 years, 65%) but also included children (21%), adolescents (11%), and aged patients (3%). One-quarter, half, and one-quarter had Venham-Score = 2, 3, and 4 at inclusion, respectively. The spatula significantly improved Oral Accessibility Score (4.8 to 10.8), Venham-Score (3.1 to 2.6), and Examiner Satisfaction Score (3.4 to 7.2) (all $p<0.001$). There were no severe spatula-related adverse events.

## Conclusion

The spatula significantly improved oral access, was safe and well-tolerated by the patients, and markedly improved oral examination quality.

## Introduction

In 2018, the World Health Organization reported that approximately 15% of the global population (1 billion people) have some form of long-term functional disability and that up to one in five of these people (110–190 million people) have a severe disability [1]. Many of these people have special healthcare needs, namely, they require medical management, healthcare interventions, and/or specialized services or programs [2].

People with special needs due to underlying disabilities experience enormous inequality in terms of oral disease prevalence and unmet oral healthcare needs [3]. These inequalities reflect in part discrimination against people with disability. In addition, oral health is often not prioritized in people with disability and few dentists are trained to manage them effectively [3, 4]. The underlying disability also contributes in various ways to the parlous oral health of disabled people: thus, people with disability may be unable to brush their teeth or explain they are in pain, their special medications may promote oral disease, their ability to clear food from the oral cavity may be lower, they may prefer carbohydrate-rich foods, and they may have lower immunity [5–11].

Another important reason for the poor oral health of people with disability is that many are anxious about and therefore resist dental treatment [10, 11]. For example, Gordon et al.[12] showed that of 494 members of a special-needs association, 55.2% reported feeling fearful about dental treatment; of these,13.2% reported feeling 'very afraid' or 'terrified' (by contrast, 40% of the general population report some dental fear and about 5% are truly phobic [13]). This anxiety and aversion is often exacerbated by chronic illness and frequent hospital visits [14]. Anxiety in this population is a significant barrier to the utilization of dental services [3, 12, 14, 15]: of the group studied by Gordon et al. [12], 17.2% reported that dental fear was an important reason for not seeking dental care, and fear associated significantly with failure to seek dental care.

Such dental anxiety can make it difficult for dental surgeons to access the oral cavity in people with physical, intellectual, or behavioral disabilities. In particular, anxiety in people with intellectual or behavioral disabilities causes them to adopt protective behaviors such as refusing to open their mouth or limiting the mouth opening. They may also find it easier, more comfortable, and reassuring to bite down [16]. People with physical/motor function disabilities may also have difficulty keeping their mouth open for a long time; in addition, they may be prone to involuntary mandibular movements [17, 18]. These behaviors pose a significant

problem for dental surgeons during oral examinations and treatments. Indeed, while many studies show that dental practitioners want to reduce the disparity between people with and without disability in terms of dental care, most find working with patients with disability challenging [19–21].

Consequently, compared to the general population, people with disability have high unmet needs for oral disease prevention and periodontal, restorative, and functional treatment [3]. As a result, they have poor oral health, as shown by significantly higher incidences of dental caries and periodontal diseases [5, 6, 22].

Since poor oral health affects esthetics, communication, behavior, self-esteem, social integration, and quality of life in people with physical, intellectual, and/or behavioral disorders [14, 23], it is essential to make it easier for dental surgeons to access the oral cavity of people with disability.

One way is to use mouth gags, retractors, props, or similar devices. Many such devices have been patented and/or are on the market [24] but it should be noted that with very rare exceptions [25, 26], none have been tested with clinical trials for efficacy, safety, or tolerability in even the general population. These devices also have limitations, especially for patients with disability. Thus, mouth gags are articulated devices whose prongs are inserted between the dental arches and mechanically spread with screws or ratchets [27–30]. These gags should only be used under anesthesia because they are made of metal and could cause dental fractures if the conscious patient clamps down hard. Common alternatives are (i) the Molt mouth gag, which is a metallic forceps-like device with rubber tubing over the areas where the teeth rest [31, 32], (ii) flexible retractors that spread the lips and/or cheeks away from the teeth [33] (*e.g.* OptraGate) [25, 26], (iii) flat strips of hard foam or flat bite blocks attached to a tongue depressor [24, 32], and (iv) McKesson-style mouth props: the latter are wedge-shaped devices made of rubber, plastic, or styrofoam that are inserted between the dental arches on one side [31, 32, 34, 35]. A recently reported McKesson-style mouth prop is Isolite, which simultaneously provides isolation, retraction, evacuation, and a light source [26, 31]. However, all of these alternatives can only be inserted after the mouth is already open and none have been formally tested in patients with disability.

To improve this situation, one of us (DA) recently invented the Oral Accessibility Spatula [36], which is a mouth prop that allows the dental practitioner to first gently prise apart the dental arches, keep them propped open, and, if the patient widens their arches excessively, hold the prop in place during the examination. This polypropylene trowel-like spatula consists of a handle with retention gouges, an angled neck bearing a depression for the thumb, and a short flat blade that is rounded at the end and has notches on its edges. The dental practitioner slides the blade between the maxillary and mandibular teeth and then rotates it 90˚ to open the jaws. The notches and the clamping action of the dental arches then maintain the position of the blade, thereby propping open the mouth (Fig 1).

The present prospective interventional non-randomized patient-self-controlled clinical trial evaluated how well the Oral Accessibility Spatula prototype improves oral accessibility in both minor and adult patients with special needs due to physical, intellectual, and/or behavioral disabilities during an oral examination by a special-care dental surgeon. Secondary objectives were to evaluate how well the patients tolerated the device, its safety, and how satisfied the dental practitioners were with the quality of the dental examination with the device.

## Methods

### Study design

This prospective multicenter interventional open-label one-way non-randomized patient-self-controlled clinical trial was performed in accordance with Good Clinical Practice guidelines

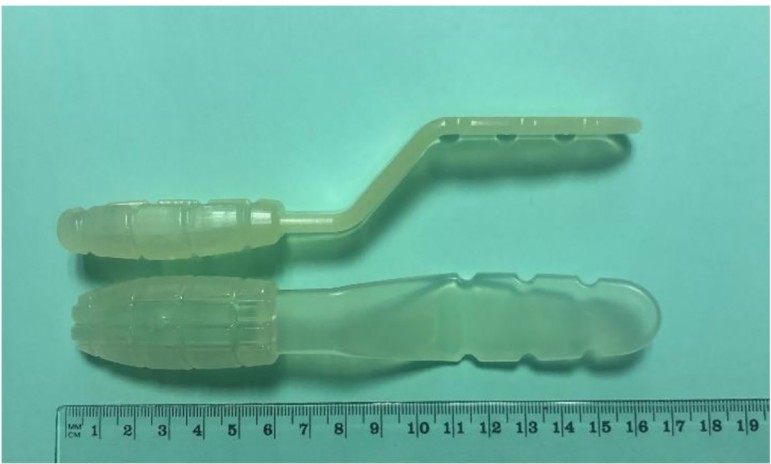

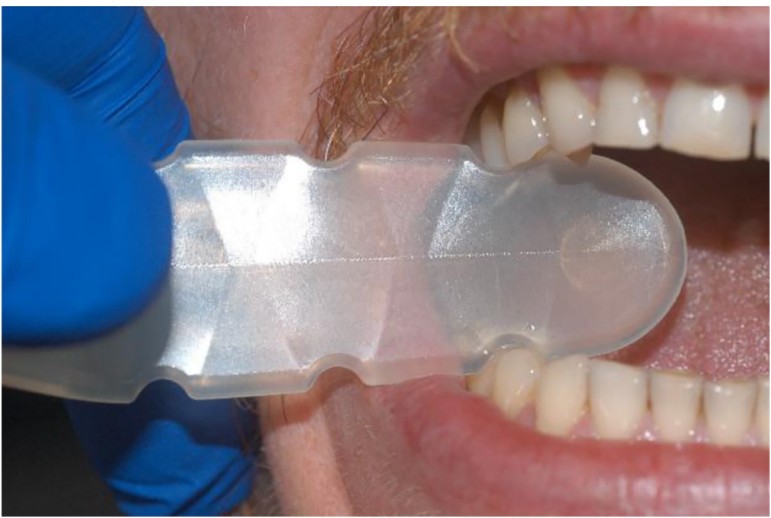

**Fig 1. The Oral Accessibility Spatula.** (Left) Photograph of the spatula. (Right) Photograph showing how the spatula is rotated after being slid between the dental arches: this causes the dental arches to separate and become wedged apart. The individual in this manuscript has given written informed consent (as outlined in PLOS consent form) to publish these case details.

and the tenets of the Declaration of Helsinki. The study protocol was approved by the CPP Est-III ethics committee and by the French regulatory agency (No. 2012-A01535-38). It was registered on ClinicalTrials.gov (No. NCT02818374) https://clinicaltrials.gov/ct2/show/results/NCT02818374?view=results. Written informed consent to participate in the study was obtained from the participants (if possible) or their legal representatives/guardians. The trial was reported according to CONSORT guidelines (S1 File).

## Participant selection

The cohort was a convenience series of minor or adult patients with special needs due to physical, intellectual, and/or behavioral disabilities who consulted a special-care dentist for a preventive check-up or an oral-dental curative act between March 30, 2016 and October 9, 2018. The disabilities were classified on the basis of the recorded clinical history of the patients. The conditions underlying the disabilities are listed in S1 Table. There were six recruitment/test

locations (Centers 01–06). All were located in the Grand-Est region of France. Five were hospital odontology outpatient departments, namely, Bel-Air Hospital (Metz-Thionville Regional Hospital) in Thionville (Center 01), Mercy Hospital (Metz-Thionville Regional Hospital) in Metz (Center 02), Brabois Hospital (Nancy Regional Teaching Hospital) in Nancy (Center 03), Civil Hospital (Strasbourg Regional Teaching Hospital) in Strasbourg (Center 04), and Emile Durkheim Hospital Center in Epinal (Center 06). The remaining Center (Center 05) was a specialized clinic, namely, St François Clinic in Haguenau.

Patient screening and inclusion depended on the modified Venham Behavioral Rating Scale, which is a validated 6-point tool that measures clinical anxiety and cooperative behavior during dental treatment [37, 38]. Scores of 0, 1, 2, 3, 4, and 5 signify states of relaxation, unease, tension, reluctance, interference, and complete refusal, respectively. Thus, potential study patients were identified during planned dental examinations, which always started in the standard manner: thus, the oral cavity was approached gradually by first touching the hand and then the lips, after which the lips were spread. If the patient had a Venham score of 0 (relaxed) or 1 (uneasy), the patient was not screened. However, if there was resistance (Venham Score ≥2), the patient was screened for inclusion in the study. Exclusion criteria were a Venham Score of 5 (completely resistant to the examination), lack of consent from the patient/the patient's representative/guardian, pregnancy or lactation, or lack of health insurance.

## The Oral Accessibility Spatula

The Oral Accessibility Spatula was designed to easily improve oral access in patients with special needs due to disability in a safe and effective manner that is tolerable for the patient. Consequently, it has an atraumatic shape. It consists of transparent Bormed ™ RF830MO polypropylene, which limits trauma to the dental elements if the spatula is bitten hard by the patient and can be easily sterilized. A patent (WO2011121513A1) was registered in 2010 [36]. The spatula is shaped like a trowel and has three rigidly attached parts. One part is a short (5 cm), flat, rigid blade with a rounded end and three pairs of symmetrical notches on its edges. The other end of the blade adjoins an angled neck bearing a thumb-shaped depression. The neck then leads to a grooved handle. The device allows the dental surgeon to spread and prop apart the patient's dental arches: thus, the blade is first inserted flat between the dental arches, after which the practitioner manually rotates the handle on a longitudinal axis, thereby moving the dental arches apart. Consequently, the blade adopts a perpendicular orientation to the dental arches and the lower and upper teeth bite into the first pair of notches near the blade tip. The remaining two notches serve to prevent inadvertent slippage of the blade into the oral cavity, which could injure the oral structures. The damage that such slippage could do is also limited by the shortness of the blade (5 cm). The blade is then held in this position by the clamping of the dental arches (or, if the patient opens their mouth wide, by the hand of the dental practitioner/assistant): there is no need to maintain great rotational force (Fig 1).

For the present study, the spatulas were single-use, sterilized by 25 kGy gamma radiation, and packaged in double individual sterile packaging. The study investigators (who were also the dental surgeons who performed the inclusion screening and the subsequent oral examinations) were trained to use the Oral Accessibility Spatula for 60 minutes before the study commenced.

## Study intervention

The study intervention consisted of two immediately successive oral examinations, first without the spatula and then with the spatula. The temporal order of the two oral examinations was fixed rather than randomized because the spatula, being a potential restraining device,

should only be used if there is difficulty obtaining access to the oral cavity. All included patients were first screened for eligibility criteria during a planned dental examination. If the patient was an autonomous adult who could provide informed consent, the intervention was performed immediately. If the patient could not provide consent, the intervention was performed only after parent/guardian authorization was obtained (up to 9 months later). Dental surgeon and/or patient blinding could not be performed because it was not possible to conceptualize a decoy device. The inventor of the spatula (DA) was sometimes an inclusion investigator but never an oral accessibility examiner. None of the other investigators had an role in the development of the spatula.

## Outcomes

The primary endpoint was the Oral Accessibility Score, which is a measure that was newly devised for this study to measure the two clinical problems that are most commonly encountered during oral examinations, namely, poor visibility and difficult dental probe examination. Simply seeing the teeth is generally not sufficient, the dental probe examination is often needed to confirm the visual results. A tool like the Oral Accessibility Score does not exist in the literature. To generate this score, the dental surgeon uses a vestibular and palatal/lingual approach and determines whether the incisor/canine, premolar, and molar tooth sectors (*i.e.* maxilla and mandible) are (i) visible and (ii) can be probed. If neither the left nor right side of the tooth sector is visible, the sector is scored 0. If either the left or right, but not both, sides of the tooth sector are visible, the sector is scored 1. If both the left and right sides of the tooth sector are visible, the sector is scored 2. The same scoring pattern is used to determine the probe-ability of the tooth sectors. These scores are then summed. Thus, the Visibility and Probe-ability Oral Accessibility Scores for each of the three tooth sectors range from 0 (the left and right sides of the tooth sector were not visible/could not be probed) to 2 (the left and right sides of the tooth sector were visible/were probed). The Total Oral Accessibility Score ranges from 0 (none of the tooth sectors were visible and could be probed) to 12 (all tooth sectors were visible and were probed). The Total Oral Acessibility Score was calculated during the examination without the spatula and again during the examination with the spatula. A Total Oral Accessibility Score of 8 was considered to indicate acceptable accessibility on the basis of clinical experience: this score means that all tooth sectors may have been seen, although they may not have been probed.

The secondary outcomes were (i) the safety of the spatula, (ii) the tolerability of the spatula for the patients, (iii) the satisfaction of the dental surgeon with the dental examination, and (iv) adverse technical events in the use of the spatula.

Spatula safety was evaluated by assessing whether it caused any of the adverse events that are frequently reported when intraoral wedges or mouthpieces are used [39]. These adverse events are labial, gingival, lingual, or cheek injuries, dental fractures, dislocations, or expulsions, nausea/vomiting, pain, joint complications, and discomfort. Thus, the patients were assessed for these adverse events starting from the beginning of the intervention through to 1 day after the intervention; however, these adverse events were only reported if they had been caused by the spatula. The patients were not assessed for these adverse events outside of this period. If these adverse events caused handicap, hospitalization, or death, they were classified as severe adverse events. The patients were also assessed for severe adverse events in general, starting from the day of inclusion in the study until 1 day after the intervention.

Spatula tolerability for the patients was assessed by applying the modified Venham Behavioral Rating Scale [37, 38] during the examination without the spatula and then again during the examination with the spatula.

Dental surgeon satisfaction was assessed by measuring the Examiner Satisfaction Scale. This self-report Likert scale was newly devised for this study and required the examiner to answer the question, "How satisfied are you in terms of the safety, duration, and quality of the examination?" The examiner provided a global score representing all three variables (safety, duration, and quality) between the numbers 0 and 10, where 0 indicated not satisfied at all and 10 indicated very satisfied. The Examiner Satisfaction Score was recorded after the examination without the spatula and then again after the examination with the spatula.

Adverse technical events in the use of the spatula were recorded by the dental surgeons after using the spatula.

## Sample size

We estimated on the basis of clinical experience that 20% of patients would achieve a Total Oral Accessibility Score of 8 when the spatula was not used. Assuming a study drop-out rate of 10%, we calculated that 200 patients would be needed to detect the ability of the spatula to increase this baseline frequency to 40% with a power of 90% and a level of significance (α) of 5%.

## Statistical analyses

Study patients with missing intervention data accounted for <2% of the eligible cohort and were excluded from analysis. The average Total Oral Accessibility Scores without and with the spatula were compared by using Wilcoxon's signed rank tests. The Visibility and Probe-ability Oral Accessibility Scores for each tooth sector without and with the spatula were also compared by using Wilcoxon's signed rank tests. The frequencies of patients with Total Oral Accessibility Scores of ≥8 were compared by using a NcNemar test. Wilcoxon's signed rank tests were also used to compare the patient Venham Scores and Examiner Satisfaction Scores without and with the spatula. Safety and adverse technical events were expressed as percentage and/or number. Logistic regression was used to identify the factors that associated with spatula-induced variation in the Total Oral Accessibility Score. Age was categorized as 0–12, 13–17, 18–65, and ≥65 years to determine whether the spatula had similar effects on oral accessibility in all age groups. Center 1 had the most patients and served as the reference site to determine whether the five other investigator sites differed in Oral Accessibility Scores. Sex, the reason for consultation, disability type, and Venham Score at inclusion were also examined for associations with Oral Accessibility Scores. The data were expressed as Odds Ratios (OR) with 95% confidence intervals (CI). The threshold of significance was 5%. All analyses were performed by using SAS/STAT software (SAS Inst., Cary, NC) version 9.4.

## Results

In total, 213 patients were assessed for eligibility between March 2016 and October 2018. Of these, nine were excluded because they declined to participate ($n = 3$) or had a Venham Score of 5 at inclusion ($n = 6$). Of the remaining 204 patients, the data of three were excluded from analysis because the patient died between the screening and intervention visits ($n = 1$) or some Oral Accessibility Score data were missing ($n = 2$). The data of the remaining 201 patients were analyzed (Fig 2).

## Baseline characteristics

The baseline clinical characteristics of the 201 patients are shown in Table 1. The patients were mostly non-elderly (18–64 years) adults ($n = 130$, 65%). The 65 juvenile patients consisted of 42 children (21% of the cohort) and 23 adolescents (11%). There were also six elderly (≥65 years) patients (3%). The male:female sex ratio was 1.1:1. Half of the patients (57%) were from

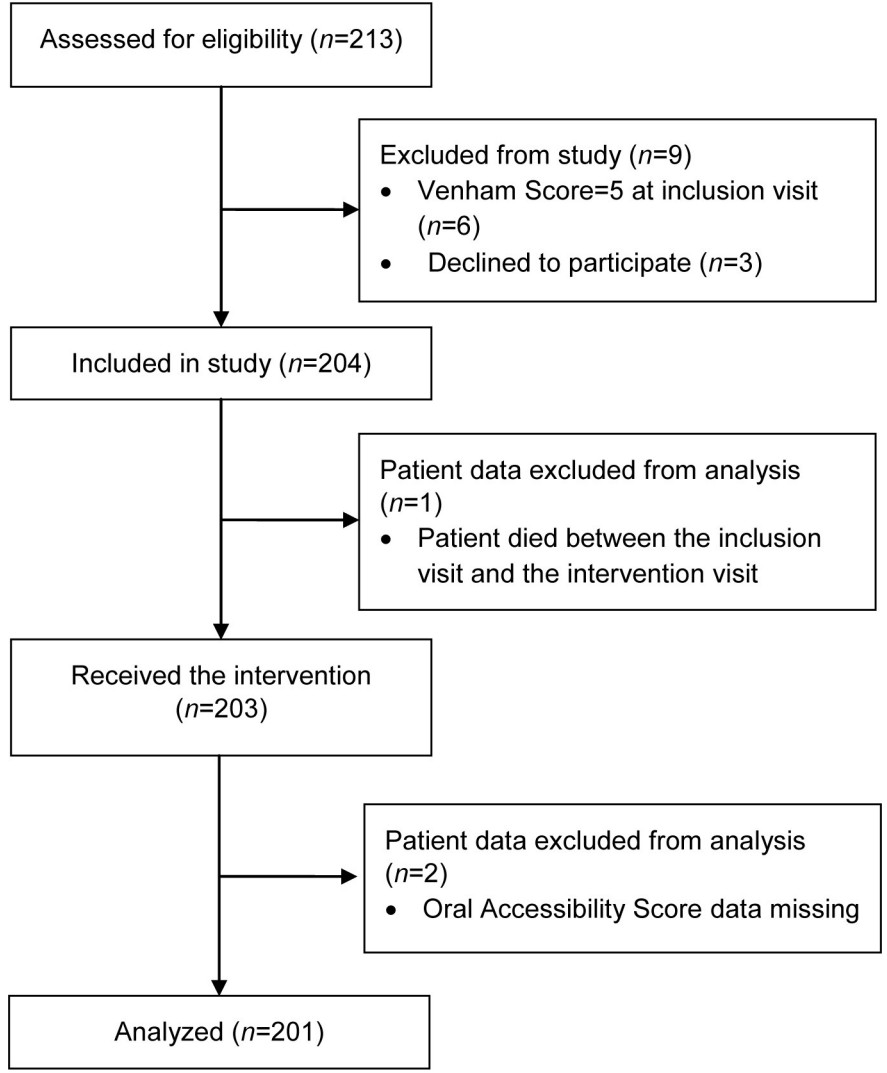

**Fig 2. Flowchart of patient disposition during the study.**

one Center (01); most of the rest were from Centers 03 (16%), 06 (12%), and 05 (9%). In terms of disability, 67% had an intellectual disability, 61% had a behavioral disability, and 43% had a physical disability. The majority of patients (80%) underwent the dental examination as a routine examination. The Venham Score of the 201 patients at inclusion was 2, 3, and 4 in 26%, 46%, and 28%, respectively. The average ± standard deviation duration between the inclusion and intervention visits was 75 ± 89 (range 0–280) days.

## Primary outcome

Use of the spatula during the oral examination asociated with significant improvement in average Total Oral Accessibility Score from 4.8±3.9 to 10.8±2.7 (maximum possible score is 12) ($p<0.001$) (Table 2). The spatula also significantly increased the frequency of patients who demonstrated acceptable accessibility (Total Oral Accessibility Score ≥8) from 25% to 91% ($p<0.001$) (Table 2).

Fig 3 shows that the spatula improved the average Visibility Oral Access Score of the three tooth sectors (incisor/canine, premolar, and molar, each with a maximum score of 2) from 1.1

**Table 1. Participant characteristics at inclusion *(n* = 201).**

| Characteristic | Mean ± SD or *n* (%) |
|---|---|
| **Age, years** | 30 ± 18 |
| 0–12 | 42 (21) |
| 13–17 | 23 (11) |
| 18–64 | 130 (65) |
| ≥65 | 6 (3) |
| **Female sex** | 95 (47) |
| **No. included in Center No.** | |
| 01 | 115 (57) |
| 02 | 7 (3) |
| 03 | 32 (16) |
| 04 | 4 (2) |
| 05 | 19 (9) |
| 06 | 24 (12) |
| **Type of disorder** | |
| physical, behavioral, and intellectual | 33 (16) |
| physical and intellectual | 29 (14) |
| physical and behavioral | 18 (9) |
| behavioral and intellectual | 37 (18) |
| physical only | 8 (4) |
| behavioral only | 37 (18) |
| Intellectual only | 39 (19) |
| **Reason for intervention consultation** | |
| Routine dental health control | 161 (80) |
| Scheduled care | 31 (15) |
| Emergency | 9 (5) |
| **Venham Score at inclusion** | |
| 2 (tense) | 52 (26) |
| 3 (reluctant) | 93 (46) |
| 4 (interference) | 56 (28) |

SD, standard deviation.

to 1.9. Similarly, the spatula improved the average Probe-ability Oral Access Score of the three tooth sectors from 0.5 to 1.7. Notably, Fig 3 also shows that these Visibility and Probe-ability Oral Accessibility Scores performed as expected when the spatula was not used. Thus, (i) all incisor/canine, premolar, and molar tooth sectors were more readily visible than they were probe-able and (ii) both visibility and probe-ability were higher for the incisors/canines than for the premolars while the molars were the least visible and probe-able of all the tooth sectors. The spatula significantly increased the visibility and probe-ability of all tooth sectors compared to when the spatula was not used (all $p<0.05$) (Fig 3).

## Secondary outcomes

The spatula associated with a significant improvement in Venham Score (from 3.1±0.8 to 2.6 ±1.0, $p<0.001$) (Table 2). The Venham Score stayed unchanged for 61% of the patients but decreased in 37% and increased in 2% (five patients).

The spatula also associated with a significant improvement in Examiner Satisfaction Score (from 3.4±2.2 to 7.2±2.1, $p<0.001$) (Table 2).

**Table 2. Effect of the spatula on primary and secondary outcome variables (_n_ = 201).**

| Variable | Examination without spatula | Examination with spatula | _P_ [a] |
|---|---|---|---|
| **Total Oral Accessibility Score (from 0 to 12)** | 4.8 ± 3.9 | 10.8 ± 2.7 | **<0.001** |
| **No. with Total Oral Accessibility Score ≥8** | 50 (25) | 183 (91) | **<0.001** |
| **Venham Behavioral Rating Score** | 3.1 ± 0.8 | 2.6 ± 1.0 | **<0.001** |
| 0: relaxed | 0 (0) | 5 (2) | |
| 1: uneasy | 3 (1) | 13 (6) | |
| 2: tense | 47 (23) | 75 (37) | |
| 3: reluctant | 84 (42) | 67 (33) | |
| 4: interference | 67 (33) | 41 (20) | |
| 5: complete refusal | 0 | 0 | |
| **Examiner Satisfaction Score (from 0 to 10)** | 3.4 ± 2.2 | 7.2 ± 2.1 | **<0.001** |

The data are shown as mean ± standard deviation or _n_ (%), as appropriate.

[a] _P_ values were determined by comparing the variable without and with the spatula by using Wilcoxon's signed rank tests or a NcNemar test.

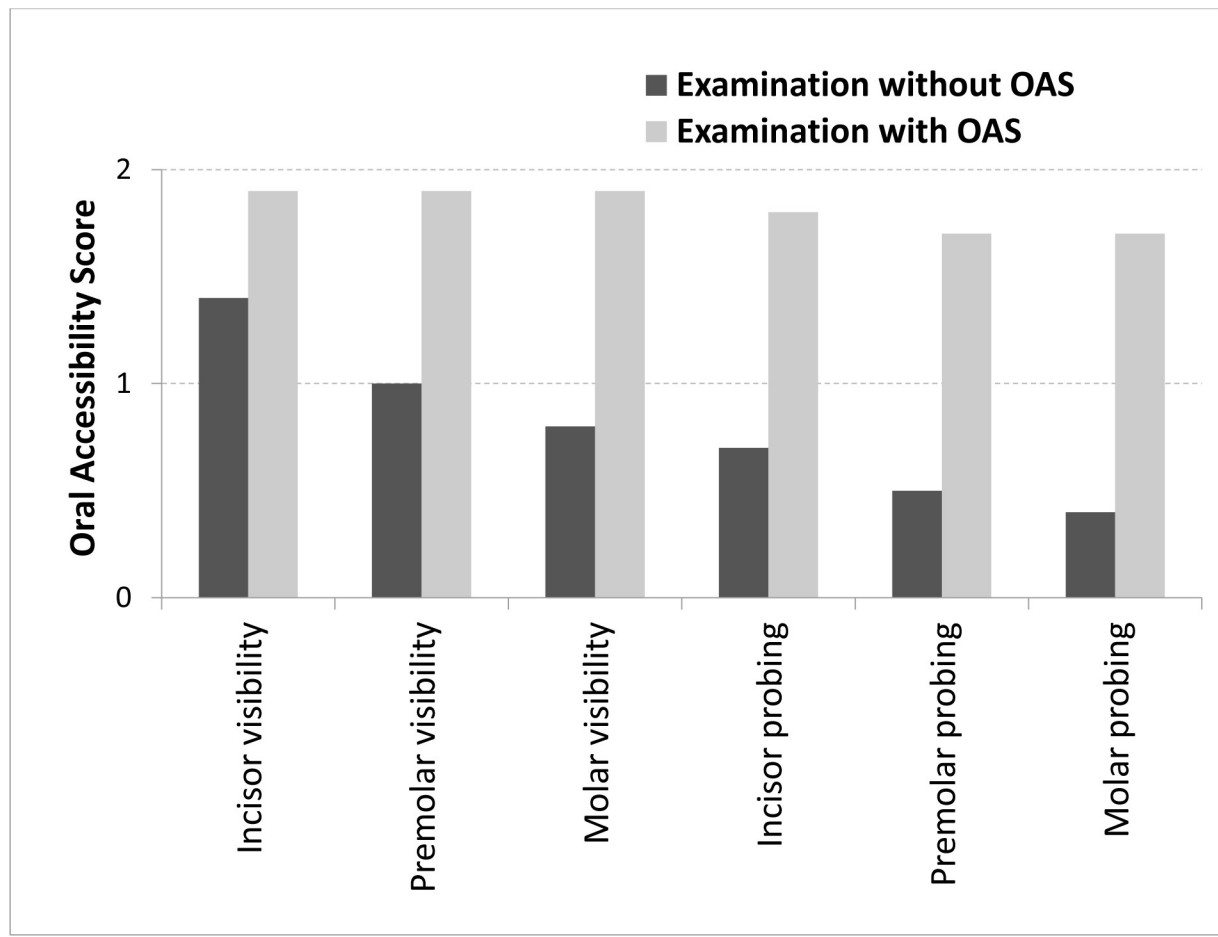

**Fig 3. Average visibility or Probe-ability oral accessibility scores for the incisor/canine, premolar, and molar tooth sectors without and with the spatula (_n_ = 201).** The score for each tooth sector was 0 if both the left and right sides of the tooth sector were not visible/probe-able; 1 if only the left or right side of the tooth sector was visible/probe-able; and 2 if both the left and right sides of the tooth sector were visible/probe-able. For all tooth sectors, the spatula significantly increased the average Visibility/Probe-ability Oral Accessibility Score (all _p_<0.05), as determined by Wilcoxon's signed ranks test. OAS, Oral Accessibility Spatula.

Table 3. Medical and technical adverse events related to the use of the spatula (*n* = 201).

| Event | *n* (% of 201 exams) |
|---|---|
| **Severe spatula-related adverse events** | 0 (0) |
| **Non-severe spatula-related adverse events** | 25 (12) |
| **Lip wound** | 6 (3) |
| **Gum wound** | 3 (1) |
| **Cheek wound** | 0 (0) |
| **Fracture of an untreated tooth** | 0 (0) |
| **Fracture of a treated tooth** | 0 (0) |
| **Dental luxation** | 0 (0) |
| **Nausea or vomiting** | 10 (5) |
| **Pain** | 5 (3) |
| **Articular complication** | 0 (0) |
| **Discomfort** | 4 (2) |
| **Adverse technical events when using the spatula** | 5 (3) |
| **Difficulty using the spatula** | 4 (2) |
| **Deformation of the spatula** | 0 (0) |
| **Rupture of the spatula** | 0 (0) |
| **Abnormal rigidity of the spatula** | 0 (0) |
| **Other: superficial scratches on the spatula** | 1 (1) |

In terms of safety, seven severe adverse events were reported during the study. None related to the use of the spatula because they occurred between the inclusion and intervention visits. One patient died and six patients were hospitalized due to preexisting pathologies. There were 25 non-severe adverse events that related to the use of the spatula and occurred during or 1 day after the examination. Nausea/vomiting was the most common (10 patients, 5% of the cohort), followed by lip wound (*n* = 6, 3%), pain (*n* = 5, 2%), discomfort (*n* = 4, 2%), and gum wound (*n* = 3, 1%). Cases of cheek wound, untreated and untreated tooth fracture, dental luxation, and articular complications were not observed (Table 3).

The dental surgeons reported five adverse technical events. There were four cases where the spatula was difficult to use because "it was impossible to verticalize the spatula due to diastema" or "too much muscular force was required". There was one case where the dental surgeon noted that the unused spatula bore superficial scratches, although they did not affect the overall structure of the tool (Table 3).

## Factors that associated with spatula-induced changes in Total Oral Accessibility Score

Table 2 shows that the spatula significantly altered the Total Oral Accessibility Score relative to the examinations without the spatula. Logistic regression then showed that the spatula significantly improved Total Oral Accessibility Score in (i) Centers 02 (OR = 0.16, 95%CI = 0.01–0.67, $p<0.001$) and 05 (OR = 0.04, 95%CI = 0.01–0.12, $p<0.001$), (ii) physically disabled patients (OR = 0.37, 95%CI = 0.22–0.64, $p = 0.001$), and (iii) patients with higher Venham Scores at inclusion (OR = 0.59, 95%CI = 0.40–0.87, $p = 0.01$). Notably, we observed significant interactions between physical disability and the other disabilities ($p = 0.01$–0.001), which suggests that if patients with a physical disability also had a behavioral and/or intellectual disability, the spatula did not improve accessibility as well as when no other disabilities were present. Relative to the unaged adult (18–64 years) group, childhood, adolescence, and an older age ($\geq$65 years) did not influence spatula-induced change in oral accessibility: thus, the spatula worked equally well with all age groups (Table 4).

**Table 4. Logistic regression analysis of factors that associated with spatula-induced change in Total Oral Accessibility Score ($n$ = 201).**

| Factor | Parameter estimate | Standard deviation | $P$ |
|---|---|---|---|
| **Age, years** | | | |
| 0–12 | 0.88 | 0.75 | 0.24 |
| 13–17 | 1.03 | 0.88 | 0.24 |
| 18–64 | - | - | - |
| ≥65 | 0.55 | 1.60 | 0.73 |
| **Sex (female *vs.* male)** | 0.46 | 0.53 | 0.39 |
| **Center No.** | | | |
| 01 | - | - | - |
| 02 | 2.05 | 1.48 | 0.17 |
| 03 | -0.80 | 0.91 | 0.38 |
| 04 | 3.55 | 1.96 | 0.07 |
| 05 | 4.78 | 0.98 | **<0.001** |
| 06 | 1.87 | 1.27 | 0.14 |
| **Reason for intervention consultation** | | | |
| Routine dental health control | - | - | - |
| Scheduled care | 0.09 | 1.08 | 0.93 |
| Emergency | 1.51 | 1.37 | 0.27 |
| **Physical disability** | 1.89 | 0.55 | **0.001** |
| **Behavioral disability** | 0.17 | 0.66 | 0.80 |
| **Intellectual disability** | -0.54 | 0.67 | 0.42 |
| **Interaction physical\*behavioral\*intellectual disability** | -4.81 | 2.09 | **0.02** |
| **Interaction physical\*behavioral disability** | -4.21 | 1.77 | **0.02** |
| **Interaction physical\*intellectual disability** | -2.96 | 1.73 | 0.09 |
| **Venham Score at inclusion** | 0.93 | 0.40 | **0.02** |

## Discussion

### Oral accessibility

The Oral Accessibility Spatula is an patented innovative device that aims to facilitate oral accessibility in patients with special needs due to physical, intellectual, and/or behavioral disorders. This prospective multicentric patient-self-controlled study showed that the device doubled the oral accessibility of these patients (Total Oral Accessibility score changed from 4.8 to 10.8; maximum possible score = 12). Notably, this improvement was observed regardless of whether the patients were children, adolescents, or older adults.

Our Oral Accessibility Scale performed as expected: when we did not use the spatula, the average score for visibility (maximum score = 2) was best for the incisors/canines (1.4), worse for the premolars (1.0), and worst of all with the molars (0.8). The same trend was observed with the average score for probe-ability (0.7, 0.5, and 0.4, respectively). The spatula so greatly improved both visibility and probe-ability that these tooth-sector accessibility trends became barely detectable (Fig 3).

### Tolerability of the spatula for the patients

Our study also showed that the spatula was well-tolerated by the patients: the spatula improved the Venham Score from 3.1 to 2.6. In fact, it improved the Venham Score in 37% of the patients and worsened it in only 2% (it did not change the Venham score in 61%).

## Safety of the spatula

It was important to study the safety of the spatula, especially in terms of oral injuries. We designed the spatula to be as safe as possible: thus, it was made from polypropylene, which we considered would be strong enough to resist mastication pressure while being soft enough to prevent tooth damage should the patient bite down hard. The blade of the spatula was also short and rounded at its end.

We observed that the spatula was safe, at least until one day after the examination: there were no severe adverse events that related to the use of the spatula. There were 25 non-severe adverse events with the spatula, with nausea/vomiting being the most common (5% of examinations). All mucous injuries were minor and did not require treatment. There were no dental injuries, dislocations, fractures, expulsions, or displacements. Articular complications were not observed. While five patients experienced pain with the spatula, the spatula did not worsen the Venham Score relative to the baseline score in any of these patients.

The safety of the spatula is further supported by the low rate of adverse technical events when using the spatula ($n = 5$). There were no cases of spatula deformation, excessive rigidity, or fracture. There were four cases where the investigator found the spatula difficult to use, either because too much muscular force was required ($n = 3$) or because the patient had a wide space between two teeth ($n = 1$). The fifth adverse technical event was that the spatula bore superficial scratches, although they did not affect the integrity of the tool.

## Dental surgeon satisfaction with the quality of the oral examination

Since oral examinations in people with special needs due to disability are often challenging [19–21], it was also important to determine how satisfied the investigators were with the examination when they used the tool. For this, we devised the self-reported Examiner Satisfaction Scale, where the investigator chose a global score between 0 and 10 that reflected the safety, duration, and quality of the examination. A score of 0 indicated the worst possible oral examination, 10 the best possible. The Examiner Satisfaction Scale was used both without and with the spatula. The spatula associated with a doubling of dental surgeon satisfaction (from 3.4 to 7.2). Thus, the examination with the spatula was overall faster and easier than without it.

## Factors that associate with improved oral accessibility

Logistic regression analysis showed that three factors associated with spatula-induced improvements in Total Oral Accessibility Score, namely, two centers (No. 02 and 05), physical disability, and higher Venham Scores at inclusion. The greater spatula-related effects in Centers 02 and 05 may reflect investigator effects that are due to low investigator numbers in these centers (two each). For example, the two investigators in Center 05 together included 19 patients. One investigator included 13 of these patients and demonstrated a striking improvement in Total Oral Accessibility Score with the spatula: the mean scores without and with the spatula were $0 \pm 0$ and $11.5 \pm 1.7$, respectively. By contrast, the mean Total Oral Accessibility Scores of the second investigator at Center No. 05 were $2.2 \pm 0.8$ and $9.0 \pm 2.3$, respectively. It is possible that the former investigator did not try strenuously to access the oral cavity without the spatula because they found access much easier to obtain with the spatula. Physical disability alone probably associated with greater spatula-induced accessibility because our physically disabled patients had cerebral palsy. The oral examination difficulties that arise with such patients generally relate to difficulties keeping their mouth open for a long time and/or involuntary mandibular movements. By contrast, patients with behavioral or intellectual disability tend to be less compliant. Indeed, we observed a significant interaction between physical disability and the other disabilities: thus, the spatula improved accessibility in patients with a physical

disability less well if they also had a behavioral and/or intellectual disability. Higher Venham Scores at inclusion may associate with better spatula-induced accessibility because there was more room for spatula-induced improvement in patients who were particularly restive at inclusion. This highlights the clinical relevance of our spatula for patients with severe disabilities that strongly limit oral accessibility.

## Ethical issues relating to the spatula

The Oral Accessibility Spatula could be perceived to be a physical restraint to be used when the patient refuses care or has to be restrained to complete the oral examination. This could produce an ethical tension in the dental surgeon that reflects their conflicting desires to simultaneously preserve patient autonomy and ultimately improve patient well-being. Indeed, dental practitioners have reported that they sometimes have to sacrifice ethical values when making a clinical decision [40]. However, many studies also show that the majority of special-care dentists accept the sparing use of limited restraint such as that involved in the use of our spatula [41]. Moreover, in our study, the spatula only worsened the Venham Score in 2% of the included patients, and only three of 213 patients (or their guardians) refused to participate in the study. These findings suggest that the patients (or guardians) did not view our spatula negatively. The latter finding is particularly striking given the fact that it can be difficult to obtain informed consent to provide a dental examination when a special-needs patient is not capable of providing that consent and therapeutic aids must be used to immobilize the patient during treatment [42].

Thus, the Oral Accessibility Spatula should be seen as an aid to improve oral accessibility rather than a constraint or physical restraint. It should always be used in a respectful manner with sensitive evaluation of the behavioral responses of the patient. It should be noted that the safety of the spatula in our study reflects the fact that it was used by special-care dentists. It is essential that dentists who are not trained to manage special-needs patients undergo training with this device. To this end, we will be producing a manual and training video for the spatula. In this setting, the Oral Accessibility Spatula can serve as a new device that facilitates preventive dental visits and dental care.

We also speculate that with appropriate training, the spatula may aid toothbrushing by caregivers, who also have difficulty accessing the oral cavity of people with disability and thus struggle to provide adequate preventive oral care [43–45]. Studies on this issue are pending.

## Study limitations

This study has a number of limitations. First, the cohort was selected by convenience sampling. Thus, it is possible that the cohort is not fully representative of the study population. Second, the same dentist performed both the examinations without and then with the spatula. It is possible that the first examination may have caused the dentist to become conversant with the patient's oral landscape, thus facilitating oral accessibility scoring on the second examination. This potential bias could have been avoided by conducting a parallel-arm trial where one arm underwent a single examination with the spatula. However, this would be undesirable because the trial design should reflect the real-life setting in which this spatula would be used, namely, it is only applied when it is needed: it is inappropriate and indeed potentially counter-productive to start an examination with a restraining device. Another possible parallel-arm trial design was one that included an arm where patients underwent two sequential examinations without a spatula. However, this design is also flawed because the second (clinically pointless) non-intervention examination could frustrate the patients and increase their resistance: the resultant loss of oral accessibility would annul the hypothetically positive effect of dentist

familiarization on Oral Accessibility Score. Moreover, the examinations with the spatula were generally faster than the examinations without it (data not shown): since longer examinations could reduce patient compliance, and a multiplicity of factors dictate examination duration in this population, it would be difficult to create second non-spatula examinations that adequately recapitulate the conditions of the spatula examination. These issues together with the vulnerability and frailty of our patient led us to exclude a second arm in the trial. In addition, the heterogeneity of the disabilities in our cohort suggests that it would be difficult to obtain a comparable group for a second arm. There is also evidence that dentist familiarization does not explain the spatula-associated improvement in Oral Accessibility Score, as follows. Thus, as indicated by our logistic analysis, higher Venham Score at inclusion associated with greater improvements in Total Oral Accessibility Score with the spatula. When we looked more closely at these data, we found that as Venham Score at inclusion rose, oral accessibility increased steadily in a 'dose-response'-like fashion (S1 Fig). Since one would anticipate dentist familiarization on the first examination to improve Total Oral Accessibility Score at the second examination to a similar degree in all patients regardless of how restive they are, it is therefore unlikely that dentist familiarization contributed significantly to the spatula-associated improvement in Oral Accessibility Score that we observed. Moreover, to overcome the potential bias of dentist familiarization, we devised the objective Oral Accessibility Score and instructed all investigators to rigorously determine this score during the examinations.

Third, it is possible that the patients became more accustomed to the dental examination process during the first examination and thus were more compliant during the second examination, thereby improving oral accessibility. This could also help explain, at least in part, why the Venham Scores generally did not worsen during the second examination. Again, this potential bias could have been avoided by conducting a parallel-arm trial where one group underwent two successive rounds of oral examination without the spatula. However, the limitations of this trial design, as discussed above, militated against adding this arm. Moreover, the majority of patients (61%) did not evince any change in Venham Score with the spatula: it seems unlikely that patient habituation was responsible for the doubling of oral accessibility when the spatula was used.

Fourth, while the sample size ($n = 201$) was sufficient to detect a significant effect of the spatula on the Total Oral Accessibility Score, there were not sufficient numbers of patients in various age groups to confirm that the spatula improved oral accessibility in all age groups: while the majority of the patients were non-elderly adults (18–64 years, $n = 130$) or children (0–12 years, $n = 42$), adolescents and especially elderly ($\geq$65 years) patients were less common ($n = 23$ and 6, respectively). Further studies with larger sample sizes in these age groups are warranted.

Fifth, the Oral Accessibility Scale and Dentist Satisfaction Score were devised for this study and have not been validated. Further studies validating these tools are needed.

This study also has positive features. In particular, our patient population came from six centers and was highly heterogeneous in terms of disabilities and age. This reflects the varied nature of the people who are seen by special-care dental surgeons. Our study showed that the spatula generally improved oral accessibility in these patients, which indicates its generalizability in this heterogeneous population. However, patients with physical disability benefitted the most from the spatula. Further studies comparing more homogeneous groups of patients with disabilities will help to indicate which patients profit the most from this tool. Finally, while many mouth gags, retractors, and props are currently on the market (or can be made from readily available materials [46]), almost none have been formally tested for clinical efficacy, safety, or tolerability: only two, OptraGate and Isolite, have been assessed by clinical trials [25, 26]. However, they were only examined for their ability to achieve tooth isolation, both

involved subjective or surrogate measures, and most importantly, neither was conducted in patients with disability. Thus, this clinical trial contributes significantly to the field of oral health in people with disability.

## Conclusion

This multicentric clinical trial showed that the Oral Accessibility Spatula was safe, significantly improved oral accessibility in patients with special needs due to disability, was tolerated well by the patients, and associated with better dental surgeon satisfaction regarding the safety, duration, and ease of the dental examination. It should be noted that the spatula that was used in this study was a prototype. Our findings suggest that this tool will help to overcome the highly prevalent obstacle of difficult oral accessibility that currently prevents patients with disability and severe disability from undergoing preventive care, especially routine dental examinations.

## Supporting information

**S1 Fig. Relationship between Venham Score at inclusion and change in visibility, probe-ability, and Total Oral Accessibility Scores when the spatula was used.** The maximum visibility and probe-ability score for each tooth sector was 2. The maximum Total Oral Accessibility Score was 12. VS, Venham Score.
(DOCX)

**S1 Table. The study patients were classified according to whether they had physical, intellectual, and/or behavioral disabilities.** The conditions underlying these disabilities are indicated below.
(DOCX)

**S1 File. CONSORT checklist.**
(DOC)

**S2 File.**
(DOC)

**S3 File.**
(DOC)

**S4 File.**
(PDF)

## Acknowledgments

The authors gratefully acknowledge the following investigators for their contributions and commitment throughout this study: Dr. Dominique Droz-Desprez and Prof. Catherine Strazielle of CHU Nancy, Dr. Amélie Dalstein of CH Emile-Durkenheim-Epinal Hospital, Dr. Sophie Albecker and Dr. Brigitte Mengus of Clinique St. François in Hagenau, Dr. Marion Strub and Prof. Marie-Cécile Manière of CHU Strasbourg, and Dr. Philippe Richardin and Dr. Stéphane Wang of CHR Metz-Thionville Mercy Hospital. We also acknowledge the contributions of Dr. Yinka Zevering of SciMeditor Scientific Writing Services to the writing of this paper.

## Author Contributions

**Conceptualization:** Daniel Anastasio.

**Data curation:** Christophe Goetz, Giuseppa Piga, Daniel Anastasio.

**Formal analysis:** Christophe Goetz.

**Funding acquisition:** Nadia Ouamara, Daniel Anastasio.

**Investigation:** Mathieu Mogenot, Laurence Hein-Halbgewachs, Dominique Droz-Desprez, Catherine Strazielle, Sylvie Albecker, Brigitte Mengus, Marion Strub, Marie-Cécile Manière, Pascal Richardin, Stéphane Wang, Giuseppa Piga, Amélie Dalstein, Daniel Anastasio.

**Methodology:** Christophe Goetz, Daniel Anastasio.

**Project administration:** Nadia Ouamara, Daniel Anastasio.

**Resources:** Christophe Goetz, Daniel Anastasio.

**Software:** Christophe Goetz.

**Supervision:** Daniel Anastasio.

**Validation:** Christophe Goetz, Daniel Anastasio.

**Visualization:** Christophe Goetz, Nadia Ouamara, Daniel Anastasio.

**Writing – original draft:** Christophe Goetz, Nadia Ouamara, Daniel Anastasio.

**Writing – review & editing:** Christophe Goetz, Nadia Ouamara, Daniel Anastasio.

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
