## [Decision Letter · Decision Letter 0]

26 Jun 2020

PONE-D-20-09935

Efficacy, tolerability, and safety of an innovative medical device for improving oral accessibility during oral examination in patients with special needs: A multicentric clinical trial

PLOS ONE

Dear Dr. Anastasio,

Thank you for submitting your manuscript to PLOS ONE. After careful consideration, we feel that it has merit but does not fully meet PLOS ONE’s publication criteria as it currently stands. Therefore, we invite you to submit a revised version of the manuscript that addresses the points raised during the review process.

We look forward to receiving your revised manuscript.

Kind regards,

Chun-Pin Lin, Ph. D.

Academic Editor

PLOS ONE

Journal Requirements:

2.

We note that you have indicated that data from this study are available upon request. PLOS only allows data to be available upon request if there are legal or ethical restrictions on sharing data publicly. For information on unacceptable data access restrictions, please see http://journals.plos.org/plosone/s/data-availability#loc-unacceptable-data-access-restrictions.

3. We note that Figure 1 includes an image of a patient / participant in the study. 

As per the PLOS ONE policy (http://journals.plos.org/plosone/s/submission-guidelines#loc-human-subjects-research) on papers that include identifying, or potentially identifying, information, the individual(s) or parent(s)/guardian(s) must be informed of the terms of the PLOS open-access (CC-BY) license and provide specific permission for publication of these details under the terms of this license.

Please download the Consent Form for Publication in a PLOS Journal (http://journals.plos.org/plosone/s/file?id=8ce6/plos-consent-form-english.pdf). The signed consent form should not be submitted with the manuscript, but should be securely filed in the individual's case notes.

Please amend the methods section and ethics statement of the manuscript to explicitly state that the patient/participant has provided consent for publication: “The individual in this manuscript has given written informed consent (as outlined in PLOS consent form) to publish these case details”.

Reviewers' comments:

Reviewer's Responses to Questions

**Comments to the Author**

1. Is the manuscript technically sound, and do the data support the conclusions?

Reviewer #1: Yes

Reviewer #2: Partly

Reviewer #3: Partly

2. Has the statistical analysis been performed appropriately and rigorously? 

Reviewer #1: Yes

Reviewer #2: Yes

Reviewer #3: Yes

3. Have the authors made all data underlying the findings in their manuscript fully available?

Reviewer #1: Yes

Reviewer #2: No

Reviewer #3: No

4. Is the manuscript presented in an intelligible fashion and written in standard English?

Reviewer #1: Yes

Reviewer #2: Yes

Reviewer #3: Yes

5. Review Comments to the Author

Reviewer #1: Important note: This review pertains only to ‘statistical aspects’ of the study and so ‘clinical aspects’ [like medical importance, relevance of the study, ‘clinical significance and implication(s)’ of the whole study, etc.] are to be evaluated [should be assessed] separately/independently. Further please note that any ‘statistical review’ is generally done under the assumption that (such) study specific methodological [as well as execution] issues are perfectly taken care of by the investigator(s). This review is not an exception to that and so does not cover clinical aspects {however, seldom comments are made only if those issues are intimately / scientifically related & intermingle with ‘statistical aspects’ of the study}.

COMMENTS:

You should have given few details [very brief] of ‘Venham test’ which, I guess, is based on eight ‘pictures’ & measures subject’s dental anxiety {maximum score 8}. As even other tools used are expected to yield data in ‘ordinal’ level of measurement, application of non-parametric test(s) [ex. Wilcoxon’s signed ranks test] was/were desirable anyway and followed here is good. However, please note that ‘Multiple linear regression’ used to identify the factors that are associated with spatula-induced variation in the ‘Total Oral Accessibility Score’ is parametric [whereas ‘logistic regression’ is semi-parametric, completely non-parametric regression is also available but not popular (& so no software available, computations are very cumbersome) and moreover, is not ‘powerful’, however, often the logistic regression is considered by many as non-parametric].

I am sure that the authors are aware of the well-known drawbacks of before-after study which is a single-arm design [a type of Quasi-experimental research, it is alright to conduct ‘single-arm’ (before-after) study when that is the only possibility]. It is very essential to keep those limitations in mind while interpreting results. Also note that a classical/ideal clinical trial/study needs/requires a concurrently handled/treated appropriated selected/chosen control/comparison parallel group/arm [pointed-out in limitations later].

I am not sure if present study be called as ‘crossover’ [line 165 in Methods - Study design section: This prospective multicenter interventional open-label one-way crossover clinical trial] when according to lines 217-220 “The study intervention consisted of two immediately successive oral examinations, first without the spatula and then with the spatula. The temporal order of the two oral examinations was fixed rather than being randomized because the spatula should only be used if there is difficulty obtaining access to the oral cavity”. If the word/term ‘one-way crossover’ clinical trial’ is planted/used here to indicate this phenomenon then it is expected to be clarified that way because the phrase ‘one-way crossover clinical trial’ is new (to most including me). When you use a word ‘crossover’ to describe a clinical trial design, a particular/typical trial design comes to mind which is entirely different than the present one.

As per account in lines 219-20 i.e. spatula should only be used if there is difficulty obtaining access to the oral cavity, shows that action later depends on outcome of first. And yet the design is called crossover. Only one variation is known [to/in the literature], called as ‘case-crossover’ design [a 'case window', the period of time during which the person was assumed to be a case, and a 'control window’ otherwise] and if that is what it meant {some similarity}, it should be clarified. If this one is proposed as new variation, please explain it in more details. {There was one paper describing/using ‘wait-listed’ control as cross-over but had given details like consequences on analyses}

It is highly appreciable that a measure called ‘Oral Accessibility Score’, which is also the primary endpoint of the study, is/ was newly devised for this study. But note that it is mandatory to study/establish its ‘psychometric properties’ [like validity, reliability] after new development / before use. Same is with the ‘Examiner Satisfaction Scale’ Likert scale which was newly devised for this study. {Note that important properties like validity, reliability incidentally is/are been called ‘psychometric properties’ but have nothing to do with ‘psychology’}.

Estimation of sample size required/needed to detect the ability of the spatula to increase 20% from baseline frequency of 20%, is correct [please make the sentence little simple to understand more clearly]. All the tools used also seems appropriately used / alright [except for properties of few].

What is the interpretation of values [parameter estimate, Standard deviation & ‘P’ for each “Factor”] displayed in table-4? Is not this a natural question in anybody’s mind? Will you please explain them little bit? Why divided (what is the purpose of dividing) ‘factor’ into categories [like Age as 0-12, 13-17, 18-64, ≥65]? According to my knowledge, nature [type & level of measurement] even of independent variables (in addition to dependent variable) matters in ‘Multiple linear regression’ [all variables are treated as numerical/continuous so that Sex (female=1, male=2) is different than (male=1, female=2)] . Why ‘Center no.1’ is taken as ‘reference category’? {this info may add to understanding, hopefully}. I guess only in logistic regression (and not in ‘Multiple linear regression’) it is necessary to identify ‘reference category’ [if there are more categories than two for a variable] because ‘odds ratio’ is estimated with reference to that category. If those reported ‘parameter estimate’s [column 2 of table 4] are ‘odds ratios’ {i.e. if binary logistic regression is used} then why not report two categories made of dependent variable (Total Oral Accessibility Score)?

Prerogative/claim mentioned in lines 400-402 [that: Consequently, it may be useful in not only dental practice but also for caregivers during toothbrushing, thereby helping to improve the parlous oral health of this population] is not clear. Will you please help me understand how it may be helping to improve the parlous oral health of this population?

Pointing-out all possible ‘limitations’ [lines 490-543; particularly highlighting that ‘potential bias could have been avoided by conducting a parallel-arm trial’ and ‘further studies validating this scale are needed’ is highly appreciated. In fact, most of the above review cover these issues only. Except these points, the article is very well drafted and study is excellently executed. Though I do not fully agree with positive features listed [lines 544-550], I can definitely say that this article should be accepted with above minor revisions suggested.

Excellent article (though non-randomized), recommending minor revision.

Reviewer #2: Introduction

In the title it is mentioned the study was conducted among special needs population. However, in the introduction the terms ‘disable’ and ‘disability’ are used. Though there is a debate if the terms special needs and disability are the same or not, it is essential the introduction should only use one term as it may confuse the reader of the context.

The term ‘special needs’ is not discussed in the introduction section. It is essential to include the prevalence, the definition, categories and relevant literature on oral access among special needs population in this section.

79 – Prevalence of anxiety among the general population and the special needs for dental procedures should be mentioned.

103-105 – The reference is on elderly population. This is not related with the special needs population

114 -140 - The description about the oral devices is too long. It should be summarised and discussed. Any previous studies done on the Efficiency, tolerability and safety of these equipment to be included

Methods

228-229 For the Oral Accessibility score – Has there been a validation study? Is there any evidence the tool is a valid one?

258-260 Has the modified Venham score been validated? Need to mention the evidence.

261 – 267 No clear description on dental surgeon satisfaction score. Need to describe if that is validated or not. Does the dental surgeon give an overall mark out of 10 or does he mark each section safety, duration and quality separately?

275-286 In the statistical analysis - Need to mention how the safety, Tolerability, satisfaction of the surgeon and adverse technical events were analysed

How the types of disability were divided into 3 segments (physical, behavioural and intellectual) components. How they were categorised needs to be mentioned.

It is clear why the randomisation and blinding were not done with the process of the study. However, it is essential to clearly mention why randomisation and blinding was not possible.

Need to mention what were the associated factors assessed clearly in the methods section.

216 – It is stated it consisted of two immediately successive oral examinations. But it seems in some cases the 2nd examination was done later. Therefore, need to correct this statement.

Results

359-363/ Table 3 - Adverse events were assessed only after one day of the intervention. In the results section it should be mentioned. Otherwise it will give a false idea to the reader.

354 – Need to explain more on the examiner satisfaction score. Clearly mention the separate scores on the different domains (safety, duration and quality)

Discussion

400 - There is no mention about the use of this device by the caregivers in introduction or methods section. Without conducting a trial on the caregivers, it is incorrect to suggest it.

422 – The adverse events were assessed only after one day of the intervention. Therefore, you can’t come to the conclusion that there were no adverse events. If you are mentioning it need to mention they were only assessed only for one day after intervention.

435 – The scoring system is discussed in this section. It needs to be included in the methodology section. Need to mention this is not validated in the limitation section

547 – There were some references about other oral devices. Need to compare this device compared to the other ones in terms of safety, tolerability and efficiency.

Need to discuss about the generalisability of the study findings including external validity and applicability

Conclusion

558-559 As this was not tested on regular toothbrushing need to remove this component

Reviewer #3: 1. The authors used the spatula to get good oral accessibility in the same participants, therefore, it is obviously that using the device could enhance the accessibility. The study did not show the unique availability. The authors did not compare the difference between using other similar devices to get oral accessibility. In this situation, it is not capable to ascertain if using other devices could also get the similar effect. The authors might find the similar device named as mouth rests or mouth stick, which also can enhance the oral accessibility. By the way, for the stick with thermoplastic rubber material on the biting site and polypropylene on the hand holding site, it is easy to handle with no or nearly few trauma for soft tissue or teeth. If there was comparison between different devices, this study could be more convincing.

2. One of the advantage of the spatula is thinner than the other devices, and the adverse events were also few in this study, however, the design of notches is possible to cause the soft tissue damage including lip and gum, especially, in using less than three notches. The retained notch without using might be one of the reasons to cause soft tissue injury.

3. The spatula seems to have the effect of tongue plate. The authors did not highlight this advantage, which is seldom found from the other devices.

4. This study had been made as well preparation, but to present a natural result. As authors’ statement, the spatula is a novel device with patent, however, the authors failed to prove the availability being better than the similar device on the market.

6. PLOS authors have the option to publish the peer review history of their article (what does this mean?). If published, this will include your full peer review and any attached files.

Reviewer #1: No

Reviewer #2: No

Reviewer #3: No

---

## [Author Response · Author response to Decision Letter 0]

5 Aug 2020

Replies to editor and reviewer comments

PONE-D-20-09935

Efficacy, tolerability, and safety of an innovative medical device for improving oral accessibility during oral examination in patients with special needs: A multicentric clinical trial

PLOS ONE

We would like to thank the three reviewers for their thorough review of our paper and their pertinent comments. We have addressed all comments to the best of our ability and feel that the paper is much improved. 

Journal Requirements:

Reply: we have ensured that the manuscript meets all PLOS One style requirements, including those for file naming.

2.

We note that you have indicated that data from this study are available upon request. PLOS only allows data to be available upon request if there are legal or ethical restrictions on sharing data publicly. For information on unacceptable data access restrictions, please see http://journals.plos.org/plosone/s/data-availability#loc-unacceptable-data-access-restrictions.

We have added the following text to the cover letter:

“(i) French national laws (loi "informatique et liberté") strictly forbid the release of data of individuals who participate in clinical studies because they may contain potentially identifying and sensitive patient information. Data requests may be sent to the corresponding author or to the data protection offficer of our hospital: dpo@chr-metz-thionville.fr”

3. We note that Figure 1 includes an image of a patient / participant in the study. 

As per the PLOS ONE policy (http://journals.plos.org/plosone/s/submission-guidelines#loc-human-subjects-research) on papers that include identifying, or potentially identifying, information, the individual(s) or parent(s)/guardian(s) must be informed of the terms of the PLOS open-access (CC-BY) license and provide specific permission for publication of these details under the terms of this license.

Please download the Consent Form for Publication in a PLOS Journal (http://journals.plos.org/plosone/s/file?id=8ce6/plos-consent-form-english.pdf). The signed consent form should not be submitted with the manuscript, but should be securely filed in the individual's case notes.

Please amend the methods section and ethics statement of the manuscript to explicitly state that the patient/participant has provided consent for publication: “The individual in this manuscript has given written informed consent (as outlined in PLOS consent form) to publish these case details”.

Reply: Since the individual in the photo was a colleague of the lead investigator (not a patient), we exchanged email correspondence with PLOS One regarding this point (23 and 24 July) and received the following reply from PLOS One on 28 July: “As your colleagues photo will be published you should seek written consent. Then please do add the following statement to the figure legend: "The individual in this manuscript has given written informed consent (as outlined in PLOS consent form) to publish these case details." In the meantime, we decided on the basis of Reviewer 3’s comments to take a new photo with a different individual that shows the position of the spatula in the mouth more clearly. We have received written informed consent from this individual and have added the above sentence to the Figure 1 legend, as suggested by PLOS One.

Comments to the Author

Note to all reviewers: We added a point to the manuscript (bolded below) that we had not emphasized in our previous version:

“Ethical issues relating to the use of the spatula: It should be noted that the safety of the spatula in our study reflects the fact that it was used by special-care dentists. It is essential that dentists who are not trained to manage special-needs patients undergo training with this device. To this end, we will be producing a manual and training video for the spatula. In this setting, the Oral Accessibility Spatula can serve as a new device that facilitates preventive dental visits and dental care. 

We also speculate that with appropriate training, the spatula may aid toothbrushing by caregivers, who also have difficulty accessing the oral cavity of people with disability and thus struggle to provide adequate preventive oral care [20–22]. Studies on this issue are pending.”

Reviewer #1: Important note: This review pertains only to ‘statistical aspects’ of the study and so ‘clinical aspects’ [like medical importance, relevance of the study, ‘clinical significance and implication(s)’ of the whole study, etc.] are to be evaluated [should be assessed] separately/independently. Further please note that any ‘statistical review’ is generally done under the assumption that (such) study specific methodological [as well as execution] issues are perfectly taken care of by the investigator(s). This review is not an exception to that and so does not cover clinical aspects {however, seldom comments are made only if those issues are intimately / scientifically related & intermingle with ‘statistical aspects’ of the study}.

COMMENTS:

You should have given few details [very brief] of ‘Venham test’ which, I guess, is based on eight ‘pictures’ & measures subject’s dental anxiety {maximum score 8}. 

Reply: We have added the following text to the Methods section: “Patient screening and inclusion depended on the modified Venham Behavioral Rating Scale, which is a validated 6-point tool that measures clinical anxiety and cooperative behavior during dental treatment [1,2]. Scores of 0, 1, 2, 3, 4, and 5 signify states of relaxation, unease, tension, reluctance, interference, and complete refusal, respectively.”

As even other tools used are expected to yield data in ‘ordinal’ level of measurement, application of non-parametric test(s) [ex. Wilcoxon’s signed ranks test] was/were desirable anyway and followed here is good. However, please note that ‘Multiple linear regression’ used to identify the factors that are associated with spatula-induced variation in the ‘Total Oral Accessibility Score’ is parametric [whereas ‘logistic regression’ is semi-parametric, completely non-parametric regression is also available but not popular (& so no software available, computations are very cumbersome) and moreover, is not ‘powerful’, however, often the logistic regression is considered by many as non-parametric].

Reply: We agree that since the Total Oral Accessibility Score data are not normally distributed, logistic regression is a more suitable test. We conducted logistic regression analysis and have therefore changed the data in Table 4. There were a few minor changes in the data findings: 

(i) Center 2, like Center 5, now displays an investigator effect. We have described these new data in the Discussion; we speculate that these strong investigator effects reflect the small number of investigators in the two centers (2 and 2, respectively).

(ii) The original interaction between spatula-induced change in oral accessibility and physical, behavioral, and intellectual disability turned from significant (p=0.02) to non-significant (p=0.053). However, the interaction with physical and intellectual disability remained significant (was p=0.02, is now p=0.001) and a new interaction with physical and behavioral disability was observed (was p=0.09, is now p=0.03). We have adjusted the Discussion slightly to include these changes but the overall picture regarding these interactions remains the same.

Higher Venham Score at inclusion remains associated with a significantly greater effect of the spatula on oral accessibility (was p=0.02, is now p=0.01).

I am sure that the authors are aware of the well-known drawbacks of before-after study which is a single-arm design [a type of Quasi-experimental research, it is alright to conduct ‘single-arm’ (before-after) study when that is the only possibility]. It is very essential to keep those limitations in mind while interpreting results. Also note that a classical/ideal clinical trial/study needs/requires a concurrently handled/treated appropriated selected/chosen control/comparison parallel group/arm [pointed-out in limitations later].

Reply: Yes, as we discussed extensively in the study limitations, a parallel arm design would certainly have been the most desirable study design. However, we decided against it for fear of placing too great a burden on this highly fragile population and because it would have been difficult to obtain a parallel arm that well recapitulated the primary arm in terms of patient disability heterogeneity and procedural variables (e.g. speed of oral examination).

I am not sure if present study be called as ‘crossover’ [line 165 in Methods - Study design section: This prospective multicenter interventional open-label one-way crossover clinical trial] when according to lines 217-220 “The study intervention consisted of two immediately successive oral examinations, first without the spatula and then with the spatula. The temporal order of the two oral examinations was fixed rather than being randomized because the spatula should only be used if there is difficulty obtaining access to the oral cavity”. If the word/term ‘one-way crossover’ clinical trial’ is planted/used here to indicate this phenomenon then it is expected to be clarified that way because the phrase ‘one-way crossover clinical trial’ is new (to most including me). When you use a word ‘crossover’ to describe a clinical trial design, a particular/typical trial design comes to mind which is entirely different than the present one.

As per account in lines 219-20 i.e. spatula should only be used if there is difficulty obtaining access to the oral cavity, shows that action later depends on outcome of first. And yet the design is called crossover. Only one variation is known [to/in the literature], called as ‘case-crossover’ design [a 'case window', the period of time during which the person was assumed to be a case, and a 'control window’ otherwise] and if that is what it meant {some similarity}, it should be clarified. If this one is proposed as new variation, please explain it in more details. {There was one paper describing/using ‘wait-listed’ control as cross-over but had given details like consequences on analyses}

Reply: We chose the term “crossover” to reflect the fact that the patients were serving as their own control. So “one-way crossover” was used to indicate the lack of randomization, which was due to the fact that oral examinations must never start with restraining devices. Nevertheless, we agree that “one-way crossover” is inappropriate because it is not a standard term and could mislead the reader. We have therefore changed the designation of the trial design to “non-randomized patient-self-controlled”.

It is highly appreciable that a measure called ‘Oral Accessibility Score’, which is also the primary endpoint of the study, is/ was newly devised for this study. But note that it is mandatory to study/establish its ‘psychometric properties’ [like validity, reliability] after new development / before use. Same is with the ‘Examiner Satisfaction Scale’ Likert scale which was newly devised for this study. {Note that important properties like validity, reliability incidentally is/are been called ‘psychometric properties’ but have nothing to do with ‘psychology’}.

Reply: The Oral Accessibility Score is more a simple metric tool rather than a psychometric tool: it consists of gathering several simple clinical observations, namely, can the dental surgeon see the teeth, yes or no? Can the dental surgeon touch the tooth/nearby gum with the probe, yes or no? As such, the Oral Accessibility Score is likely to be limited by floor or ceiling effects, or poor reproducibility or sensitivity to change, rather than a divergence between the score and the reality of oral accessibility. In other words, when we use the Oral Accessibility Score, we are more likely to risk not observing differences in score between examinations with or without the spatula rather than wrongly observing differences that do not exist in reality. 

Moreover, as we indicated in the paper, the Oral Accessibility Scale performed as expected: when we did not use the spatula, the average score for visibility (maximum score=2) was best for the incisors/canines (1.4), worse for the premolars (1.0), and worst of all with the molars (0.8). The same trend was observed with the average score for probe-ability (0.7, 0.5, and 0.4, respectively).

Nevertheless, we acknowledge that the tool should be validated and have indicated that the lack of validation is a study limitation.

Estimation of sample size required/needed to detect the ability of the spatula to increase 20% from baseline frequency of 20%, is correct [please make the sentence little simple to understand more clearly]. All the tools used also seems appropriately used / alright [except for properties of few].

Reply: We have described the sample calculation more clearly.

What is the interpretation of values [parameter estimate, Standard deviation & ‘P’ for each “Factor”] displayed in table-4? Is not this a natural question in anybody’s mind? Will you please explain them little bit? Why divided (what is the purpose of dividing) ‘factor’ into categories [like Age as 0-12, 13-17, 18-64, ≥65]? According to my knowledge, nature [type & level of measurement] even of independent variables (in addition to dependent variable) matters in ‘Multiple linear regression’ [all variables are treated as numerical/continuous so that Sex (female=1, male=2) is different than (male=1, female=2)] . Why ‘Center no.1’ is taken as ‘reference category’? {this info may add to understanding, hopefully}. I guess only in logistic regression (and not in ‘Multiple linear regression’) it is necessary to identify ‘reference category’ [if there are more categories than two for a variable] because ‘odds ratio’ is estimated with reference to that category. If those reported ‘parameter estimate’s [column 2 of table 4] are ‘odds ratios’ {i.e. if binary logistic regression is used} then why not report two categories made of dependent variable (Total Oral Accessibility Score)?

Reply: Since we have replaced the multivariate linear regression analysis with logistic regression, some of these questions are not relevant anymore. 

With regard to the question “Why divided (what is the purpose of dividing) ‘factor’ into categories [like Age as 0-12, 13-17, 18-64, ≥65]?”, we chose the four age categories because we wanted to explore whether the spatula improves oral accessibility in young children, teenagers, adults, and older adults: this is because each age group has particular issues with regard to oral accessibility. Text has been added to the Statistics section to describe this.

With regard to the question “Why ‘Center no.1’ is taken as ‘reference category’?”, we chose Center 1 because that is the center in which the lead study investigator works and it had the most patients; we wanted to see whether there were site-related variations in terms of how much the spatula improved oral accessibility. Text has been added to the Statistics section to describe this.

Prerogative/claim mentioned in lines 400-402 [that: Consequently, it may be useful in not only dental practice but also for caregivers during toothbrushing, thereby helping to improve the parlous oral health of this population] is not clear. Will you please help me understand how it may be helping to improve the parlous oral health of this population?

Reply: Many people who have a physical, intellectual, and/or behavioral disability (including aged people) cannot adequately perform daily tasks such as toothbrushing and require a caregiver (e.g. a relative or nursing aide). As was described in the Introduction, the task of toothbrushing can be very difficult in this population due to their disabilities and/or high levels of anxiety. A large issue is that the caregiver cannot access the oral cavity well or long enough to successfully complete effective toothbrushing. As a result, people with a disability often have very poor oral health that impacts on many aspects of their lives. The spatula is a simple tool that we suspect will also greatly help caregivers to brush the teeth of their charges. We are currently planning a trial to test this hypothesis but we had wanted to indicate in the paper that it is likely that this tool may have wider usefulness in the special-needs community than just aiding dental surgeon activities.

However, another reviewer correctly pointed out that our study did not test the efficacy of the spatula in the hands of caregivers and that mentioning caregivers was confusing. We agree and have therefore removed all mention of caregivers from the Introduction and only briefly mentioned the possibility that the spatula might be able to help caregivers as well at the end of the Ethical issues section in the Discussion.

Pointing-out all possible ‘limitations’ [lines 490-543; particularly highlighting that ‘potential bias could have been avoided by conducting a parallel-arm trial’ and ‘further studies validating this scale are needed’ is highly appreciated. In fact, most of the above review cover these issues only. Except these points, the article is very well drafted and study is excellently executed. Though I do not fully agree with positive features listed [lines 544-550], I can definitely say that this article should be accepted with above minor revisions suggested.

Excellent article (though non-randomized), recommending minor revision.

Reply: Thank you. We are grateful for your thorough comments and feel they have improved our paper.

Reviewer #2: Introduction

In the title it is mentioned the study was conducted among special needs population. However, in the introduction the terms ‘disable’ and ‘disability’ are used. Though there is a debate if the terms special needs and disability are the same or not, it is essential the introduction should only use one term as it may confuse the reader of the context.

The term ‘special needs’ is not discussed in the introduction section. 

Reply: We agree that the terms “disability” and “special needs” are not exactly the same. We have therefore changed the Introduction to more clearly define these terms, as follows (bolded). The patients were thereafter mostly referred to as “patients with special needs due to physical, intellectual, and/or behavioral disabilities”: 

“In 2018, the World Health Organization reported that approximately 15% of the global population (1 billion people) have some form of long-term functional disability and that up to one in five of these people (110–190 million people) have a severe disability [1]. Many of these people have special healthcare needs, namely, they require medical management, healthcare interventions, and/or specialized services or programs [2]”

Reference 2 is a formal and recent (2016) definition of the term “special needs” by the American Academy of Pediatric Dentisty: “Special health care needs include any physical, developmental, mental, sensory, behavioral, cognitive, or emotional impairment or limiting condition that requires medical management, health care intervention, and/or use of specialized services or programs.” https://www.aapd.org/media/Policies_Guidelines/D_SHCN.pdf

It is essential to include the prevalence, the definition, categories and relevant literature on oral access among special needs population in this section.

Reply: In the literature to date, oral access (in both the general and special-needs populations) is examined via surrogate measures such as the frequency of caries or other oral diseases.

As far as we know, our study is the first to provide a quantitative definition of oral accessibility: we defined acceptable access as an Oral Accessibility Score of 8. It should be noted that the Oral Accessibility Score was devised for our study because to our knowledge, there are no other such scores in the literature. 

79 – Prevalence of anxiety among the general population and the special needs for dental procedures should be mentioned.

Reply: Anxiety in the special-needs population is rather poorly studied and to our knowledge, a study directly comparing populations with and without special needs in terms of the prevalence of dental anxiety has not been performed. 

We adapted the Introduction text as follows (bolded text):

“Another important reason for the poor oral health of people with disability is that many are anxious about and therefore resist dental treatment [9,10]. For example, Gordon et al.[12] showed that of 494 members of a special-needs association, 55.2% reported feeling fearful about dental treatment; of these,13.2% reported feeling ‘very afraid’ or ‘terrified’ (by contrast, 40% of the general population report some dental fear and about 5% are truly phobic [13]).

103-105 – The reference is on elderly population. This is not related with the special needs population

Reply: We deleted this reference.

114 -140 - The description about the oral devices is too long. It should be summarised and discussed. Any previous studies done on the Efficiency, tolerability and safety of these equipment to be included

Reply: The oral device summary in the Introduction was reduced.

In terms of other studies, we made an exhaustive search both before the study and while writing this paper for other clinical studies on oral accessibility devices but only found two. Both are on commercial devices. One was by Jawa et al. (reference 25 in the paper). They conducted a randomized crossover study in 30 healthy 6–8-year-old children that compared a conventional bite block to a new oral device called OptraGate. The study endpoints were apparently subjective opinions by the two operators regarding the ability to maintain isolation and patient cooperation. The conclusion was that OptraGate was better than the bite block in terms of both variables.

The second study was by Alhareky et al. (reference 26 in the paper). They conducted a randomized split-face study with 42 7–16-year-old healthy patients. Isolation was performed on one (randomized) side with a conventional rubber dam while the other side was isolated with a device called Isolite. After isolation, fissure sealant was applied to the isolated teeth. The operators determined how long it took to isolate the teeth and apply the sealant. Patient satisfaction was determined with a questionnaire. Isolite associated with shorter chair time and greater patient satisfaction.

Thus, neither study (i) directly quantitated the effect of the device on oral accessibility with a scale, (ii) examined the safety of the device, (iii) quantitated the tolerability of the device with a validated scale, and (iv) most importantly, examined the efficacy of the device in patients with disability.

We believe that the present study is thus novel in the special-care dentistry field.

Methods

228-229 For the Oral Accessibility score – Has there been a validation study? Is there any evidence the tool is a valid one?

Reply: The Oral Accessibility Score has not been validated. 

We would like to point out that the Oral Accessibility Score is a simple metric tool (rather than a psychometric tool): it consists of gathering several simple clinical observations, namely, does the dental surgeon see the teeth, yes or no? Can the dental surgeon touch the tooth/nearby gum with the probe, yes or no? As such, the Oral Accessibility Score is likely to be limited by floor or ceiling effects, or poor reproducibility or sensitivity to change, rather than a divergence between the score and the reality of oral accessibility. In other words, when we use the Oral Accessibility Score, we are more likely to risk not observing differences in score between examinations with or without the spatula rather than wrongly observing differences that do not exist in reality. 

Moreover, as described in the paper, the Oral Accessibility Scale performed as expected: when we did not use the spatula, the average score for visibility (maximum score=2) was best for the incisors/canines (1.4), worse for the premolars (1.0), and worst of all with the molars (0.8). The same trend was observed with the average score for probe-ability (0.7, 0.5, and 0.4, respectively).

Nonetheless, we recognize that the tool should be validated and have acknowledged this lack in the study limitations section.

258-260 Has the modified Venham score been validated? Need to mention the evidence.

Reply: Yes, the Venham Score has been validated by its creator (reference 37). The fact that it has been validated has been added to the Methods section.

261 – 267 No clear description on dental surgeon satisfaction score. Need to describe if that is validated or not. Does the dental surgeon give an overall mark out of 10 or does he mark each section safety, duration and quality separately?

Reply: The Dentist Satisfaction Score was also not validated. It is a subjective score that is based on a self-report Likert scale. It consists of the question, "How satisfied are you in terms of the safety, duration, and quality of the examination?” The examiner provides a score between the numbers 0 and 10, where 0 indicates not satisfied at all and 10 indicates very satisfied.

Thus, safety, duration, and quality are measured globally rather than individually. The Methods section was changed as follows to make this clear: “Dental surgeon satisfaction was assessed by measuring the Examiner Satisfaction Scale. This self-report Likert scale was newly devised for this study and required the examiner to answer the question, "How satisfied are you in terms of the safety, duration, and quality of the examination?” The examiner provided a global score representing all three variables (safety, duration, and quality) between the numbers 0 and 10”

The fact that this score has not been validated has been added to the study limitations section.

275-286 In the statistical analysis - Need to mention how the safety, Tolerability, satisfaction of the surgeon and adverse technical events were analysed

Reply: With regard to tolerability and surgeon satisfaction, the original manuscript stated “Wilcoxon's signed rank tests were also used to compare the patient Venham Scores and Examiner Satisfaction Scores without and with the spatula.” 

With regard to safety and adverse technical events, we added the following text to the statistical analysis section: “Safety and adverse technical events were expressed as percentage and/or number.”

How the types of disability were divided into 3 segments (physical, behavioural and intellectual) components. How they were categorised needs to be mentioned.

Reply: We addressed this as follows (bolded text and supplementary table):

“Participant selection: The cohort was a convenience series of minor or adult patients with special needs due to physical, intellectual, and/or behavioral disabilities who consulted a special-care dentist for a preventive check-up or an oral-dental curative act between March 30, 2016 and October 9, 2018. The disabilities were classified on the basis of the recorded clinical history of the patients. The conditions underlying the disabilities are listed in Table S1.”

Table S1 The study patients were classified according to whether they had intellectual, physical, and/or behavioral disabilities. The conditions underlying these disabilities are indicated below 

Conditions associated with physical disability Conditions associated with intellectual disability Conditions associated with behavioral disability

Stroke sequelae Stroke sequelae Richardson’s syndrome

Drowning sequelae Alzheimer’s disease Alzheimer’s disease

Cerebral palsy Cerebral palsy Cerebral palsy

Blindness Autism spectrum disorder Autism spectrum disorder

Congenital encephalopathy Congenital encephalopathy Dravet syndrome

Polymalformative syndromes and associated anomalies Polymalformative syndromes and associated anomalies Polymalformative syndromes and associated anomalies

Richardson’s disease Lewy body dementia Congenital encephalopathy

Huntington's chorea Microcephaly Huntington's chorea

Monosomy 7 Monosomy 7 Monosomy 7

Car accident sequelae Psychosis Psychosis

Tuberous sclerosis Joubert syndrome Tuberous sclerosis

Angelman syndrome Angelman syndrome Angelman syndrome

Goldenhar syndrome Goldenhar syndrome CASK-related disorders

Meningitis sequelae Kleefstra syndrome Recklinghausen’s disease

Ruptured aneurysm sequelae Martin-Bell syndrome Martin-Bell syndrome

Rett syndrome Rett syndrome Goldenhar syndrome

Trisomy 18,20,22 Morning glory syndrome Trisomy 18,20,22

Down’s syndrome Down’s syndrome Mowat Wilson syndrome

Trisomy 5 Trisomy 5 

It is clear why the randomisation and blinding were not done with the process of the study. However, it is essential to clearly mention why randomisation and blinding was not possible.

Reply: With regard to randomization, we have adapted the original text in the Methods: “The temporal order of the two oral examinations was fixed rather than being randomized because the spatula, being a potential restraining device, should only be used if there is difficulty obtaining access to the oral cavity.”

With regard to blinding, we have added the following text to the Methods section: “Dental surgeon and/or patient blinding could not be performed because it was not possible to conceptualize a decoy device.”

Need to mention what were the associated factors assessed clearly in the methods section.

Reply: The following text has been added to the statistics section (note that we changed the multivariate linear regression to logistic regression on the advice of another reviewer): “Logistic regression was used to identify the factors that associated with spatula-induced variation in the Total Oral Accessibility Score. Age was categorized as 0–12, 13–17, 18–65, and ≥65 years to determine whether the spatula had similar effects on oral accessibility in all age groups. Center 1 had the most patients and served as the reference site to determine whether the five other investigator sites differed in Oral Accessibility Scores. Sex, the reason for consultation, disability type, and Venham Score at inclusion were also examined for associations with Oral Accessibility Scores. The data were expressed as Odds Ratios (OR) with 95% confidence intervals (CI).”

216 – It is stated it consisted of two immediately successive oral examinations. But it seems in some cases the 2nd examination was done later. Therefore, need to correct this statement.

Reply: In all patients, the examination without the spatula was succeeded immediately by the examination with the spatula.

There may have been some confusion with the fact that for some patients, there was a time interval between the screening/inclusion visit and the examination visit (the examination visit consisted of the immediately successive examinations without and then with the spatula). The original Methods stated: “All included patients were first screened for eligibility criteria during a planned dental examination. If the patient was an autonomous adult who could provide informed consent, the intervention was performed immediately. If the patient could not provide consent, the intervention was performed only after parent/guardian authorization was obtained (up to 9 months later).” 

The original Results stated: “The average ± standard deviation duration between the inclusion and intervention visits was 75 ± 89 (range 0–280) days.”

Results

359-363/ Table 3 - Adverse events were assessed only after one day of the intervention. In the results section it should be mentioned. Otherwise it will give a false idea to the reader.

Reply: We have adapted the Results section as follows: “There were 25 non-severe adverse events that related to the use of the spatula and occurred during or 1 day after the examination”

354 – Need to explain more on the examiner satisfaction score. Clearly mention the separate scores on the different domains (safety, duration and quality)

Reply: Safety, duration, and quality were not measured individually. The Methods section was changed as follows to make this clear: “Dental surgeon satisfaction was assessed by measuring the Examiner Satisfaction Scale. This self-report Likert scale was newly devised for this study and required the examiner to answer the question, "How satisfied are you in terms of the safety, duration, and quality of the examination?” The examiner provided a global score representing all three variables (safety, duration, and quality) between the numbers 0 and 10”

Discussion

400 - There is no mention about the use of this device by the caregivers in introduction or methods section. Without conducting a trial on the caregivers, it is incorrect to suggest it.

Reply: Indeed, the study was on dental surgeons only. We are planning a trial on caregivers because we anticipate that the spatula may be a boon for caregivers as well. We wanted to indicate the likelihood that the spatula will also be useful to others working with the special needs community. However, we agree with your point. We eliminated all mention of caregivers from the Introduction and modified the Discussion so that there is only a short section on caregivers at the end of the ‘Ethical issues relating to the spatula’ section:

“…the Oral Accessibility Spatula can serve as a new device that facilitates preventive dental visits and dental care. 

We also speculate that the spatula may aid toothbrushing by caregivers, who also have difficulty accessing the oral cavity of people with disability and sometimes develop negative attitudes towards providing preventive oral care to people with disability [20–23]. Studies on this issue are pending.”

422 – The adverse events were assessed only after one day of the intervention. Therefore, you can’t come to the conclusion that there were no adverse events. If you are mentioning it need to mention they were only assessed only for one day after intervention.

Reply: We adapted the Discussion text as follows: “We observed that the spatula was safe, at least until one day after the examination”

435 – The scoring system is discussed in this section. It needs to be included in the methodology section. Need to mention this is not validated in the limitation section

Reply: Thank you for this point. We moved the Discussion text about the Oral Accessibility Score to the Methods section.

The fact that this tool has not been validated was stated in the original manuscript: “Fifth, the Oral Accessibility Scale was devised for this study and has not been validated.” 

547 – There were some references about other oral devices. Need to compare this device compared to the other ones in terms of safety, tolerability and efficiency.

Reply: As we indicated in response to your comments on the Introduction above, there are only two clinical studies on oral devices, neither of which (i) directly quantitated the effect of the device on oral accessibility with a scale, (ii) examined the safety of the device, (iii) quantitated the tolerability of the device with a validated scale, and (iv) most importantly, examined the efficacy of the device in patients with disability.

We believe that the present study is thus novel in the special-needs dentistry field.

We have sought to make this point more clearly in the Discussion: “Finally, while many mouth gags, retractors, and props are currently on the market (or can be made from readily available materials [7]), almost none have been formally tested for clinical efficacy, safety, or tolerability: only two, OptraGate and Isolite, have been assessed by clinical trials [35,41]. However, they were only examined for their ability to achieve tooth isolation, both involved subjective or surrogate measures, and most importantly, neither was conducted in patients with disability.”

We also made this point clearer in the Introduction:

“One way is to use mouth gags, retractors, props, or similar devices. Many such devices have been patented and/or are on the market [26] but it should be noted that with very rare exceptions [35,41], none have been tested with clinical trials for efficacy, safety, or tolerability in even the general population. These devices also have limitations, especially for patients with disability. Thus, mouth gags are… However, all of these alternatives can only be inserted after the mouth is already open and none have been formally tested in patients with disability”

Need to discuss about the generalisability of the study findings including external validity and applicability

Reply: We added the following text to the end of the Discussion:

“This study also has positive features. In particular, our patient population came from six centers and was highly heterogeneous in terms of disabilities and age. This reflects the varied nature of the people who are seen by special-care dental surgeons. Our study showed that the spatula generally improved oral accessibility in these patients, which indicates its generalizability in this heterogeneous population. However, patients with physical disability benefitted the most from the spatula. Further studies comparing more homogeneous groups of patients with disabilities will help to indicate which patients profit the most from this tool.”

Conclusion

558-559 As this was not tested on regular toothbrushing need to remove this component

Reply: We agree and deleted the text on toothbrushing.

Thank you very much for your comments. They have helped us to improve our manuscript.

Reviewer #3: 1. The authors used the spatula to get good oral accessibility in the same participants, therefore, it is obviously that using the device could enhance the accessibility. The study did not show the unique availability. The authors did not compare the difference between using other similar devices to get oral accessibility. In this situation, it is not capable to ascertain if using other devices could also get the similar effect. The authors might find the similar device named as mouth rests or mouth stick, which also can enhance the oral accessibility. By the way, for the stick with thermoplastic rubber material on the biting site and polypropylene on the hand holding site, it is easy to handle with no or nearly few trauma for soft tissue or teeth. If there was comparison between different devices, this study could be more convincing.

Reply: It was not possible to employ a comparator arm with another oral accessibility device because as far as we know, there are no suitable oral accessibility devices that have been validated with a clinical trial in a special-needs population. 

We made an exhaustive search both before the study and while writing this paper for other clinical studies on oral accessibility devices but only found two. Both are on commercial devices. One was by Jawa et al. (reference 25 in the paper). They conducted a randomized crossover study in 30 healthy 6–8-year-old children that compared a conventional bite block to a new isolation device called OptraGate. The study endpoints were apparently subjective opinions by the two operators regarding the ability to maintain isolation and patient cooperation. The conclusion was that OptraGate was better than the bite block in terms of both variables.

The second study was by Alhareky et al. (reference 26 in the paper). They conducted a randomized split-face study with 42 7–16-year-old healthy patients. Isolation was performed on one (randomized) side with a conventional rubber dam while the other side was isolated with a device called Isolite. After isolation, fissure sealant was applied to the isolated teeth. The operators determined how long it took to isolate the teeth and apply the sealant. Patient satisfaction was determined with a questionnaire. Isolite associated with shorter chair time and greater patient satisfaction.

Thus, neither study (i) directly quantitated the effect of the device on oral accessibility with a scale, (ii) examined the safety of the device, (iii) quantitated the tolerability of the device with a validated scale, and (iv) most importantly, examined the efficacy of the device in patients with disability. 

In addition, both Isolite and OptraGate are used for isolation, not simple oral examinations. Isolite simultaneously provides a bite block, isolation, suction, and retraction but is a large, complex device that may intimidate special-needs patients, many of whom already have high levels of anxiety about dental examinations. It also requires the patient to open their mouth widely to insert the bite block. The OptraGate mouth retractor requires an additional step of fitting the retractor between the entire lips and gums of the patient. This step would increase the examination time and is likely to further agitate special-needs patients. It also requires patient cooperation to place it and will not stop the teeth from closing under force. Thus, neither device could serve as a comparator ‘gold standard’ device in our trial.

The device you mentioned, namely, a stick with thermoplastic rubber material on the biting site and polypropylene on the hand-holding site, seems to be a home-made device? Homemade devices are not suitable for clinical studies because their composition may vary, which could yield variation in efficacy and adverse technical and safety events. In any case, we have not found any clinical trials on such a device. If there are such trials, we would be grateful if you could provide the reference(s).

We thus believe that the present study is novel in the special-care dentistry field.

2. One of the advantage of the spatula is thinner than the other devices, and the adverse events were also few in this study, however, the design of notches is possible to cause the soft tissue damage including lip and gum, especially, in using less than three notches. The retained notch without using might be one of the reasons to cause soft tissue injury.

Reply: Thank you for this pertinent point, it made us realize that our picture of the use of the spatula was misleading. The notches on the spatula are actually safety features: they are there merely to stop the spatula from slipping too far into the oral cavity, perhaps during a restive movement by the patient. Without the notches, the blade of the spatula could inadvertently be thrust into the oral structures, thus causing injury. The spatula is generally always used with the teeth resting on the first pair of notches (i.e. the notches closest to the tip of the blade). The second and third pairs of notches mainly* serve as back-ups – if the spatula slips over the first pair of notches, the second and certainly the third pair will arrest the blade from going in further. 

It is for this reason too that the blade is only 5-cm long: this is long enough to have some wiggle room and short enough that the blade will never be able to do serious injury in the case that the blade slips to the third pair of notches. 

* Very occasionally, the second pair of notches are used to obtain a slightly wider mouth opening.

We have changed the photo to better reflect the standard use of the spatula (see below). We have also added some texts to the manuscript to emphasize these points. We feel that it is unlikely that the unused notches could cause soft tissue injuries because as you can see from the photo below, all notches are bevelled and rounded and lack sharp edges.

3. The spatula seems to have the effect of tongue plate. The authors did not highlight this advantage, which is seldom found from the other devices.

Reply: The spatula in its standard use does not act as a tongue plate. This can be seen from the new photo: when the teeth rest on the first pair of notches, the tongue is clearly not depressed. The blade is also too short (5-cm long) to have any serious tongue-depressive activity even when the teeth rest on the third pair of notches (which they generally would not). 

4. This study had been made as well preparation, but to present a natural result. As authors’ statement, the spatula is a novel device with patent, however, the authors failed to prove the availability being better than the similar device on the market.

Reply: As we stated in our reponse to your first comment, similar devices on the market have not been tested for efficacy, safety, and tolerability by clinical trials. The present study is in fact the first to test a factory-produced oral accessibility device for its efficacy, safety, and tolerability in special-needs patients.

Thank you very much for your comments. They have helped us to improve our manuscript.

---

## [Decision Letter · Decision Letter 1]

16 Sep 2020

Efficacy, tolerability, and safety of an innovative medical device for improving oral accessibility during oral examination in special-needs patients: A multicentric clinical trial

PONE-D-20-09935R1

Dear Dr. Anastasio,

We’re pleased to inform you that your manuscript has been judged scientifically suitable for publication and will be formally accepted for publication once it meets all outstanding technical requirements.

Kind regards,

Chun-Pin Lin, Ph. D.

Academic Editor

PLOS ONE

Additional Editor Comments (optional):

Reviewers' comments:

Reviewer's Responses to Questions

**Comments to the Author**

1. If the authors have adequately addressed your comments raised in a previous round of review and you feel that this manuscript is now acceptable for publication, you may indicate that here to bypass the “Comments to the Author” section, enter your conflict of interest statement in the “Confidential to Editor” section, and submit your "Accept" recommendation.

Reviewer #1: All comments have been addressed

Reviewer #2: All comments have been addressed

2. Is the manuscript technically sound, and do the data support the conclusions?

Reviewer #1: Yes

Reviewer #2: Yes

3. Has the statistical analysis been performed appropriately and rigorously? 

Reviewer #1: Yes

Reviewer #2: Yes

4. Have the authors made all data underlying the findings in their manuscript fully available?

Reviewer #1: Yes

Reviewer #2: No

5. Is the manuscript presented in an intelligible fashion and written in standard English?

Reviewer #1: Yes

Reviewer #2: Yes

6. Review Comments to the Author

Reviewer #1: COMMENTS: Since the comments made on earlier draft by me (and hopefully by other respected reviewers also) are attended positively/adequately, I am satisfied and, in my opinion, the manuscript is improved a lot. I recommend acceptance, as I commented earlier that the article is very well drafted and study is excellently executed, and now it has achieved acceptable level of our journal. Please accept.

Reviewer #2: All the comments given were addressed by the authors and it seems we can accept this article for publication

7. PLOS authors have the option to publish the peer review history of their article (what does this mean?). If published, this will include your full peer review and any attached files.

Reviewer #1: **Yes: **Dr. Sanjeev Sarmukaddam

Reviewer #2: **Yes: **Dr. Mahesh Kumbukage (MBBS, MSc)

---

## [Editor Report · Acceptance letter]

18 Sep 2020

PONE-D-20-09935R1 

Efficacy, tolerability, and safety of an innovative medical device for improving oral accessibility during oral examination in special-needs patients: A multicentric clinical trial 

Dear Dr. Anastasio:

I'm pleased to inform you that your manuscript has been deemed suitable for publication in PLOS ONE. Congratulations! Your manuscript is now with our production department. 

Kind regards, 

on behalf of

Dr. Chun-Pin Lin 

Academic Editor

PLOS ONE